# Single cell transcriptional signatures of the human placenta in term and preterm parturition

Roger Pique-Regi[1,2,3]*, Roberto Romero[1,3,4,5,6]*, Adi L Tarca[2,3,7], Edward D Sendler[1], Yi Xu[2,3], Valeria Garcia-Flores[2,3], Yaozhu Leng[2,3], Francesca Luca[1,2], Sonia S Hassan[2,8], Nardhy Gomez-Lopez[2,3,9]*

[1]Center for Molecular Medicine and Genetics, Wayne State University, Detroit, United States; [2]Department of Obstetrics and Gynecology, Wayne State University School of Medicine, Detroit, United States; [3]Perinatology Research Branch, Division of Obstetrics and Maternal-Fetal Medicine, Division of Intramural Research, *Eunice Kennedy Shriver* National Institute of Child Health and Human Development, National Institutes of Health, U.S. Department of Health and Human Services, Detroit, United States; [4]Department of Obstetrics and Gynecology, University of Michigan, Ann Arbor, United States; [5]Department of Epidemiology and Biostatistics, Michigan State University, East Lansing, United States; [6]Detroit Medical Center, Detroit, United States; [7]Department of Computer Science, College of Engineering, Wayne State University, Detroit, United States; [8]Department of Physiology, Wayne State University School of Medicine, Detroit, United States; [9]Department of Immunology, Microbiology, and Biochemistry, Wayne State University School of Medicine, Detroit, United States

**\*For correspondence:**
rpique@wayne.edu (RP-R);
prbchiefstaff@med.wayne.edu
(RR);
ngomezlo@med.wayne.edu (NG-L)

**Competing interests:** The authors declare that no competing interests exist.

**Abstract** More than 135 million births occur each year; yet, the molecular underpinnings of human parturition in gestational tissues, and in particular the placenta, are still poorly understood. The placenta is a complex heterogeneous organ including cells of both maternal and fetal origin, and insults that disrupt the maternal-fetal dialogue could result in adverse pregnancy outcomes such as preterm birth. There is limited knowledge of the cell type composition and transcriptional activity of the placenta and its compartments during physiologic and pathologic parturition. To fill this knowledge gap, we used scRNA-seq to profile the placental villous tree, basal plate, and chorioamniotic membranes of women with or without labor at term and those with preterm labor. Significant differences in cell type composition and transcriptional profiles were found among placental compartments and across study groups. For the first time, two cell types were identified: 1) lymphatic endothelial decidual cells in the chorioamniotic membranes, and 2) non-proliferative interstitial cytotrophoblasts in the placental villi. Maternal macrophages from the chorioamniotic membranes displayed the largest differences in gene expression (e.g. *NFKB1*) in both processes of labor; yet, specific gene expression changes were also detected in preterm labor. Importantly, several placental scRNA-seq transcriptional signatures were modulated with advancing gestation in the maternal circulation, and specific immune cell type signatures were increased with labor at term (NK-cell and activated T-cell signatures) and with preterm labor (macrophage, monocyte, and activated T-cell signatures). Herein, we provide a catalogue of cell types and transcriptional profiles in the human placenta, shedding light on the molecular underpinnings and non-invasive prediction of the physiologic and pathologic parturition.

## Introduction

Parturition is essential for the reproductive success of viviparous species (*Romero et al., 2006a*); yet, the mechanisms responsible for the onset of labor remain to be elucidated (*Norwitz et al., 1999*; *Norwitz et al., 2015*). Understanding human parturition is essential to tackle the challenge of prematurity, which affects 15 million neonates every year (*Muglia and Katz, 2010*; *Blencowe et al., 2012*; *Romero et al., 2014a*). Bulk transcriptomic studies of the cervix (*Hassan et al., 2006*; *Hassan et al., 2007*; *Hassan et al., 2009*; *Bollopragada et al., 2009*; *Dobyns et al., 2015*), myometrium (*Charpigny et al., 2003*; *Romero et al., 2014b*; *Mittal et al., 2010*; *Mittal et al., 2011*; *Chan et al., 2014*; *Stanfield et al., 2019*), and chorioamniotic membranes (*Haddad et al., 2006*; *Mittal et al., 2009*; *Nhan-Chang et al., 2010*) revealed that labor is a state of physiological inflammation; however, finding specific pathways implicated in preterm labor still remains an elusive goal. A possible explanation is that gestational tissues, and especially the placenta, are heterogeneous composites of multiple cell types, and elucidating perturbations in the maternal-fetal dialogue requires dissection of the transcriptional activity at the cell type level, which is not possible using bulk analyses. Recent microfluidic and droplet-based technological advances have enabled characterization of gene expression at single-cell resolution (scRNA-seq) (*Klein et al., 2015*; *Macosko et al., 2015*). Previous work in humans (*Tsang et al., 2017*; *Pavličev et al., 2017*; *Vento-Tormo et al., 2018*) and mice (*Nelson et al., 2016*) demonstrated that scRNA-seq can capture the multiple cell types that constitute the placenta and identify their maternal or fetal origin. Such studies showed that single-cell technology can be used to infer communication networks across the different cell types at the maternal-fetal interface (*Vento-Tormo et al., 2018*). Further, the single-cell-derived placental signatures were detected in the cell-free RNA present in maternal circulation (*Tsang et al., 2017*), suggesting that non-invasive identification of women with early-onset pre-eclampsia is feasible. However, these studies included a limited number of samples and did not account for the fact that different pathologies can arise from dysfunction in different placental compartments. In addition, the physiologic and pathologic processes of labor have never been studied at single-cell resolution.

## Results and discussion

In this study, a total of 25 scRNA-seq libraries were prepared from three placental compartments: basal plate (BP), placental villous (PV), and chorioamniotic membranes (CAM) (*Figure 1A*). These were collected from nine women in the following study groups: term no labor (TNL), term in labor (TIL), and preterm labor (PTL). scRNA-seq libraries were prepared with the 10X Chromium system and were processed using the 10X Cell Ranger software, resulting in 79,906 cells being captured and profiled across all samples (*Supplementary file 1*). We used Seurat (*Butler et al., 2018*) to normalize expression profiles and identified 19 distinct clusters, which were assigned to cell types based on the expression of previously reported marker genes (*Tsang et al., 2017*; *Pavličev et al., 2017*; *Vento-Tormo et al., 2018*) (see Materials and methods, *Figure 1—figure supplement 1* and *Supplementary file 2–3*). The uniform manifold approximation and projection (UMAP *Becht et al., 2019*) was used to display these clusters in two dimensions (*Figure 1B*). With this approach, the local and global topological structure of the clusters is preserved, with subtypes of the major cell lineages (trophoblast, lymphoid, myeloid, stromal, and endothelial sub-clusters) being displayed proximal to each other. The trophoblast lineage reconstruction displayed in *Figure 1—figure supplement 2* shows the progression from cytotrophoblasts to either extravillous trophoblasts or syncytiotrophoblasts, which recapitulates the differentiation structure previously reported (*Tsang et al., 2017*; *Vento-Tormo et al., 2018*).

The cell type composition differed both among placental compartments (*Figure 1C*) and due to the presence of physiologic and pathologic processes of labor (i.e. term in labor and preterm labor) (*Figure 1D*). While extravillous trophoblasts (EVT) were present in all three compartments, cytotrophoblasts (CTB) were especially pervasive in the placental villi, which is explained by the fact that CTBs are abundant in the parenchyma of the placentas. CTBs were also present in the basal plate since this placental compartment is adjacent to the placental villi (*Figure 1A*). The phenotypic similarities between trophoblasts in proximity to the decidua parietalis (layer attached to the chorioamniotic membranes) and those found in the basal plate have been previously documented



**Figure 1.** Transcriptional map of the placenta in human parturition. (**A**) Study design illustrating the placental compartments and study groups. (**B**) Uniform Manifold Approximation Plot (UMAP), where dots represent single cells and are colored by cell type. (**C**) Distribution of single-cell clusters by placental compartments. (**D**) Average proportions of cell types by placental compartments and study groups. (**E**) Distribution of single cells by maternal or fetal origin. STB, Syncytiotrophoblast; EVT, Extravillous trophoblast; CTB, cytotrophoblast; HSC, hematopoietic stem cell; npiCTB, non proliferative interstitial cytotrophoblast; LED, lymphoid endothelial decidual cell.

The online version of this article includes the following figure supplement(s) for figure 1:

**Figure supplement 1.** Heatmap of the top gene expression markers defining each cell-type.
**Figure supplement 2.** UMAP plot highlighting the trophoblast cell-types and their inferred differentiation path using slingshot R package.
**Figure supplement 3.** Single marker gene expression UMAP plot for genes differentially expressed between CTB and npiCTB.
**Figure supplement 4.** Analysis of the fetal/maternal origin of the cell-types based on data from three pregnancies with a male fetus.
**Figure supplement 5.** Alluvial diagram showing the correspondence between our final curated cluster labels and automated cell-labeling methods.
*Figure 1 continued on next page*

**Figure supplement 6.** Heatmap showing the correspondence between our final curated cluster labels and automated cell-labeling methods.

**Figure supplement 7.** Uniform Manifold Approximation Plot (UMAP), where dots representing single cells and color represents Seurat predicted cell type labels.

**Figure supplement 8.** Doublet analysis by DoubletFinder.

(*Genbačev et al., 2015*; *Garrido-Gomez et al., 2017*). Importantly, non-proliferative interstitial cyto-trophoblasts (npiCTB) were identified for the first time in the placental villi as forming a distinct cluster. This new cluster was also observed in the basal plate, but not in the chorioamniotic membranes, suggesting that this type of trophoblast has specific functions in the placental tree. Lineage reconstruction by slingshot (*Street et al., 2018*) revealed that npiCTBs are likely derived from conventional CTBs (*Figure 1—figure supplement 2*). The non-proliferative nature of npiCTBs was evidenced by the reduced expression of genes involved in cell proliferation such as *XIST*, *DDX3X*, and *EIF1AX* (*Figure 1—figure supplement 3*). npiCTBs displayed an increased expression of *PAGE4* (*Figure 1—figure supplement 3*), a gene expressed by CTBs isolated from pregnancy terminations (*Genbacev et al., 2011*), suggesting that this type of trophoblast cell is present earlier in gestation. As expected, trophoblast cell types were of fetal origin, and decidual cells present in the basal plate (including the decidua basalis) and chorioamniotic membranes (including the decidua parietalis) were of maternal origin (*Figure 1E* and *Figure 1—figure supplement 4*).

In terms of immune cell types, the chorioamniotic membranes largely contained lymphoid and myeloid cells of maternal origin, including T cells (mostly in a resting state), NK cells, and macrophages (*Figure 1C and E* and *Figure 1—figure supplement 4*). In contrast, the basal plate included immune cells of both maternal and fetal origin, such as T cells (mostly in an active state), NK cells, and macrophages. The placental villi contained more fetal than maternal immune cells, namely monocytes, macrophages, T cells, and NK cells. Two macrophage subsets were found in placenta compartments: macrophage 1 of maternal origin that was predominant in the chorioamniotic membranes, and macrophage 2 of fetal origin that was mainly present in the basal plate and placental villi. Together with previous single cell studies of early pregnancy (*Vento-Tormo et al., 2018*), these results highlight the complexity and dynamics of the immune cellular composition of the placental tissues, including the maternal-fetal interface (i.e. decidua), from early gestation to term or preterm delivery.

Importantly, a new lymphatic endothelial decidual (LED) cell type of maternal origin was identified in the chorioamniotic membranes, forming a distinct transcriptional cluster that was separate from other endothelial cell-types (*Figure 1C and E*). LED cells were rarely observed in the basal plate and were completely absent in the placental villous. Similar to other endothelial cell types, LED cells highly expressed *CD34*, *CDH5*, *EDNRB*, *PDPN*, and *TIE1* (*Figure 2—figure supplement 1*). The signature genes of this novel cell type were enriched for pathways involving cell-cell and cell-surface interactions at the vascular wall, extracellular matrix organization (*Figure 2—figure supplement 2*), tight junction, and focal adhesion (*Figure 2—figure supplement 3*), indicating that LEDs possess the machinery required to mediate the influx of immune cells into the chorioamniotic membranes. Immunostaining confirmed the co-expression of LYVE1 (lymphatic marker) and CD31 (endothelial molecule marker) in the vessels of the decidua parietalis of the chorioamniotic membranes, but not in the basal plate or placenta (*Figure 2A*). The co-localization of LYVE1 and CD31 proteins (i.e. LED cells) in the chorioamniotic membranes is shown in *Figure 2B* and *Figure 2—video 1*. LED cells also expressed the common endothelial cell marker *CD34* (*Figure 2C*, green arrow). *LYVE1* was also expressed by the fetal macrophages present in the placental villi and basal plate (*Figure 2C*, red arrow), yet the protein was only visualized by immunostaining in immune cells located in the villous tree (*Figure 2A*, red arrows). This finding conclusively shows the presence of lymphatic vessels in the decidua parietalis of the chorioamniotic membranes, providing a major route for maternal lymphocytes (e.g. T cells) infiltrating the maternal-fetal interface (*Arenas-Hernandez et al., 2019*).

For cell types that were present in more than one placental compartment, major differences in gene expression were identified across locations, indicative of further specialization of cells depending on the unique physiological functions of each microenvironment (*Figure 3—figure supplement 1* and *Supplementary file 4*). Differences in the transcriptional profiles were particularly large for



**Figure 2.** Identification of LED cells in the chorioamniotic membranes. (**A**) Cell segmentation map (built using the DAPI nuclear staining) and immunofluorescence detection of LYVE-1 (red) and CD31 (green) in the basal plate (BP), placental villi (PV), and chorioamniotic membranes (CAM). Red arrows point to fetal macrophages expressing LYVE1 but not CD31, and green arrows indicate lymphatic endothelial decidual cells (LED cells)
*Figure 2 continued on next page*

*Figure 2 continued*

expressing both LYVE1 and CD31. (B) Co-expression of LYVE1 and CD31 (i.e. LED cells) in the chorioamniotic membranes. (C) Single-cell expression UMAP of LYVE-1 (red) and CD34 (green) in the placental compartments.

The online version of this article includes the following video and figure supplement(s) for figure 2:

**Figure supplement 1.** Single marker gene expression UMAP plot for genes that are more highly expressed in lymphatic endothelial decidual (LED) cells.

**Figure supplement 2.** Clusterprofiler dot plot showing the ReactomeDB Pathways enriched for genes that define each cell-type.

**Figure supplement 3.** Clusterprofiler dot plot showing the Kegg Pathways enriched for genes that define each cell-type.

**Figure 2—video 1.** Video with the 3D reconstruction of the lymphatic endothelium in the decidua present in the CAM compartment.

https://elifesciences.org/articles/52004#fig2video1

maternal macrophages as well as EVTs, NK cells, and T cells in the chorioamniotic membranes compared to the other compartments. Genes differentially expressed in the chorioamniotic membranes were enriched for interleukin and Toll-like receptor signaling as well as for the NF-κB and TNF pathways (*Figure 3—figure supplements 2–4*). These results are consistent with previous reports showing a role for these mediators in the inflammatory process of labor (*Romero et al., 1989a*; *Romero et al., 1990b*; *Romero et al., 1992a*; *Romero et al., 1990a*; *Santhanam et al., 1991*; *Romero et al., 1993*; *Romero et al., 1991*; *Hsu et al., 1998*; *Keelan et al., 1999*; *Young et al., 2002*; *Osman et al., 2003*; *Kim et al., 2004*; *Abrahams et al., 2004*; *Kumazaki et al., 2004*; *Koga et al., 2009*; *Belt et al., 1999*; *Yan et al., 2002*; *Lindström and Bennett, 2005*; *Vora et al., 2010*; *Romero, 1989b*; *Romero et al., 1992b*; *Lonergan et al., 2003*). Conversely, the placental villous and basal plate were more similar to each other, with most differentially expressed genes (DEG) between these compartments being noted in fibroblasts (335 DEG, q < 0.1 and fold change >2) (*Figure 3—figure supplements 1* and *5–10*). DEGs in the placental villous fibroblasts showed enrichment in smooth muscle contraction, the apelin and oxytocin signaling pathways (*Figure 3—figure supplement 9*), while DEGs in CAM fibroblasts were enriched in elastic fiber formation and extracellular matrix pathways (*Figure 3—figure supplement 2*). The latter finding indicates that the same cell type (e.g. fibroblasts) may have distinct functions in different microenvironments of the placenta.

Next, we assessed changes due to term and preterm labor in each cell type (*Supplementary file 5*). The largest number of DEGs between the term labor and term no labor groups were observed in the maternal macrophages (macrophage 1), followed by the EVT (144 and 37, respectively, q < 0.1; *Figure 3A*). The largest number of DEGs between the preterm labor and term labor groups were observed in EVT and CTB (37 and 33, respectively, q < 0.1; *Figure 3A*). *Figure 3B* displays the gene expression changes between TIL and TNL or PTL and TNL that are shared between the two labor groups, representing the common pathway of parturition (defined as the anatomical, physiological, biochemical, endocrinological, immunological, and clinical events that occur in the mother and/or fetus in both term and preterm labor *Romero et al., 2006b*). Non-shared differences in gene expression with labor at term and in preterm labor were mostly observed in trophoblast cell types such as CTB and EVT as well as in stromal cells (*Figure 3C*). Some of these changes may be explained by the unavoidable confounding effect of gestational age since placentas from women without labor in preterm gestation cannot be obtained in the absence of pregnancy complications. Specifically, the expression of *NFKB1* by maternal macrophages was higher in women with term labor compared to non-labor controls, and this increase was further accentuated in preterm labor (*Figure 3D*). Consistent with the induction of the NFκB pathway, the labor-associated DEGs in macrophages involved biological processes such as activation of immune response and regulation of pro-inflammatory cytokine production (*Figure 3—figure supplement 11A*). These results are in line with previous studies showing that decidual macrophages undergo an M1-like macrophage polarization (i.e. pro-inflammatory phenotype) during term and preterm labor (*Xu et al., 2016*).

When comparing the effect sizes between the PTL/TNL and TIL/TNL juxtapositions on the same gene and cell type, positive correlations were observed for most of the placental cell types (*Figure 3E*). Genes displaying differential effects in term and preterm labor are mostly found in trophoblast cell types (see off-diagonal points in the scatter plot), which may be explained by the phenomenon of gene expression decoherence (*Lea et al., 2019*). This lack of proper correlation between biomarkers to their expected normal relationships is commonly found in pathological

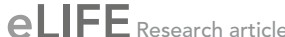

**Figure 3.** Cell type specific expression changes in term and preterm labor. (**A**) Number of differentially expressed genes (DEGs) among study groups (TNL, term no labor; TIL, term in labor; PTL, preterm labor) by direction of change. Shared (**B**) and non-shared (**C**) expression changes in term labor and preterm labor relative to the term no labor group (q < 0.01). The length of each whisker represents the 95% confidence interval. (**D**) The expression of *NFKB1* by maternal macrophages in the placental compartments (BP, basal plate; PV, placental villous; CAM, chorioamniotic membranes) and study groups. The notch represents the 95% confidence interval of the median. (**E**) Differences and similarities in expression changes with preterm labor and term labor by three major cell types (immune, stromal/endothelial, and trophoblast cells).

The online version of this article includes the following figure supplement(s) for figure 3:

**Figure supplement 1.** Stacked bar plot summarizing differentially expressed genes across compartments for a cell types that are present on all three of them.

**Figure supplement 2.** Clusterprofiler dot plot showing the ReactomeDB Pathways enriched for genes that are significantly more highly expressed in the CAM compartment relative to the other compartments for each cell-type.

**Figure supplement 3.** Clusterprofiler dot plot showing the Kegg Pathways enriched for genes that are significantly more highly expressed in the CAM compartment relative to the other compartments for each cell-type.

**Figure supplement 4.** Clusterprofiler dot plot showing gene ontology (GO) terms enriched for genes that are significantly more highly expressed in the CAM compartment relative to the other compartments for each cell-type.

**Figure supplement 5.** Clusterprofiler dot plot showing the ReactomeDB Pathways enriched for genes that are significantly more highly expressed in the BP compartment relative to the other compartments for each cell-type.

**Figure supplement 6.** Clusterprofiler dot plot showing the Kegg Pathways enriched for genes that are significantly more highly expressed in the BP compartment relative to the other compartments for each cell-type.

**Figure supplement 7.** Clusterprofiler dot plot showing gene ontology (GO) terms enriched for genes that are significantly more highly expressed in the BP compartment relative to the other compartments for each cell-type.

**Figure supplement 8.** Clusterprofiler dot plot showing the ReactomeDB Pathways enriched for genes that are significantly more highly expressed in the PV compartment relative to the other compartments for each cell-type.

*Figure 3 continued on next page*

*Figure 3 continued*

**Figure supplement 9.** Clusterprofiler dot plot showing the Kegg Pathways enriched for genes that are significantly more highly expressed in the PV compartment relative to the other compartments for each cell-type.

**Figure supplement 10.** Clusterprofiler dot plot showing gene ontology (GO) terms enriched for genes that are significantly more highly expressed in the PV compartment relative to the other compartments for each cell-type.

**Figure supplement 11.** Clusterprofiler dot plot showing ReactomeDB pathways enriched using gene set enrichment analysis (GSEA) for genes differentially expressed in term labor compared to term no labor condition.

conditions. Lastly, in EVT the DEGs with labor were enriched for genes implicated in cellular response to stress, including the WNT and NOTCH pathways, as well as cell cycle checkpoints (*Figure 3—figure supplement 11B*), further supporting the hypothesis that the cellular senescence pathway (i.e. cell cycle arrest) is implicated in the physiologic (*Behnia et al., 2015*; *Polettini et al., 2015*) and pathologic (*Hirota et al., 2010*; *Gomez-Lopez et al., 2017*) processes of labor.

To demonstrate the translational value of single-cell RNA signatures derived from the placenta, we conducted an in silico analysis in public datasets (*Tarca et al., 2019*; *Paquette et al., 2018*) to test whether the single-cell signatures could be non-invasively monitored in the maternal circulation throughout gestation (*Figure 4A*). Previous studies have correlated bulk mRNA expression in the maternal circulation with gestational age at blood draw (*Tarca et al., 2019*; *Al-Garawi et al., 2016*), risk for preterm birth (*Paquette et al., 2018*; *Heng et al., 2014*; *Sirota et al., 2018*; *Knijnenburg et al., 2019*), or both (*Heng et al., 2016*; *Ngo et al., 2018*). First, using whole blood bulk RNAseq data, we quantified the expression of single-cell signatures in the maternal circulation. We found that the expression of the single-cell signatures of macrophages, monocytes, NK cells, T cells, npiCTB, and fibroblasts is modulated with advancing gestational age (*Figure 4B–C*, *Figure 4—figure supplement 1A*). These results validate the T-cell and monocyte signature changes with gestational age that were previously reported (*Tsang et al., 2017*; *Tarca et al., 2019*); yet, here we show that novel placental single-cell signatures (e.g., npiCTB and fibroblast) can also be non-invasively monitored in maternal circulation (*Figure 4—figure supplement 1A*). In addition, for the first time, we report that the expression of the single-cell signatures of NK-cells and activated T-cells were upregulated in women with spontaneous labor at term compared to gestational-age matched controls without labor (*Figure 4D*). Furthermore, we found that the average expression of the single-cell signatures of macrophages, monocytes, activated T cells, and fibroblasts were increased in the circulation of women with preterm labor and delivery compared to gestational age-matched controls (24–34 weeks of gestation) (*Figure 4E* and *Figure 4—figure supplement 1B*). These findings are in line with previous reports indicating a role for these immune cell types in the pathophysiology of preterm labor (*Arenas-Hernandez et al., 2019*; *Hamilton et al., 2012*; *Shynlova et al., 2013*; *Gomez-Lopez et al., 2016*).

## Conclusion

In summary, this study provides evidence of differences in cell type composition and transcriptional profiles among the basal plate, placental villi, and chorioamniotic membranes, as well as between the pathologic and physiologic processes of labor at single-cell resolution. Using scRNAseq technology, two novel cell types were identified in the chorioamniotic membranes and placental villi. In addition, we showed that maternal macrophages and extravillous trophoblasts are the cell types with the most transcriptional changes during the process of labor. Importantly, many of the genes differentially expressed in these cell-types replicate for both conditions of labor. This result shows that we have enough statistical power to detect the changes in gene expression with a large effect size that are general or a common molecular pathway in parturition; yet, additional studies are needed to characterize the different etiologies of the preterm labor syndrome. Lastly, we report that maternal and fetal transcriptional signatures derived from placental scRNA-seq are modulated with advancing gestation and are markedly perturbed with term and preterm labor in the maternal circulation. These results highlight the potential of single-cell signatures as biomarkers to non-invasively monitor the cellular dynamics during pregnancy and to predict obstetrical disease. The current study represents the most comprehensive single-cell analysis of the human placental transcriptome in physiologic and pathologic parturition.

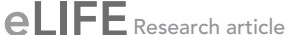

**Figure 4.** In silico analysis to quantify scRNA-seq signatures in the maternal circulation. (**A**) Diagram of the longitudinal study used to generate bulk RNAseq data (GSE114037) (*Tarca et al., 2019*) to evaluate changes in scRNA-seq signatures with advancing gestation. Whole blood samples were collected throughout gestation from women who delivered at term. (**B and C**) Variation of scRNA-seq signature expression in the maternal circulation with advancing gestation. (**D**) Diagram of the cross-sectional study used to generate bulk RNAseq data (GSE114037) to evaluate changes in scRNA-seq

*Figure 4 continued*

signatures with labor at term (*Tarca et al., 2019*). Differences in the expression of scRNA-seq signatures between women with spontaneous labor at term (TIL) and term no labor controls (TNL). (E) Diagram of the cross-sectional study used to generate bulk RNAseq data (GSE96083) to evaluate changes in scRNA-seq signatures in preterm labor (*Paquette et al., 2018*). Differences in the expression of scRNA-seq signatures between women with spontaneous preterm labor (PTL) and gestational-age matched controls (GA control).

The online version of this article includes the following figure supplement(s) for figure 4:

**Figure supplement 1.** Quantification of scRNA-seq signatures in maternal circulation (continued from main *Figure 4*).

## Materials and methods

### Sample collection and processing, single-cell preparation, library preparation, and sequencing

#### Human subjects

Immediately after delivery, placental samples [the villi, basal plate (including the decidua basalis) and chorioamniotic membranes (including the decidua parietalis)] were collected from women with or without labor at term or preterm labor at the Detroit Medical Center, Wayne State University School of Medicine (Detroit, MI). Labor was defined by the presence of regular uterine contractions at a frequency of at least two contractions every 10 min with cervical changes resulting in delivery. Women with preterm labor delivered between 33–35 weeks of gestation whereas those with term labor delivered between 38–40 weeks of gestation (*Supplementary file 6*). The collection and use of human materials for research purposes were approved by the Institutional Review Boards of the Wayne State University School of Medicine. All participating women provided written informed consent prior to sample collection.

#### Single-cell preparation

Cells from placental villi, basal plate, and chorioamniotic membranes were isolated by enzymatic digestion, using previously described protocols with modifications (*Tsang et al., 2017*; *Xu et al., 2015*). Briefly, placental tissues were homogenized using a gentleMACS Dissociator (Miltenyi Biotec, San Diego, CA) either in an enzyme cocktail from the Umbilical Cord Dissociation Kit (Miltenyi Biotec) or in collagenase A (Sigma Aldrich, St. Louis, MO). After digestion, homogenized tissues were washed with ice-cold 1X phosphate-buffered saline (PBS) and filtered through a cell strainer (Fisher Scientific, Durham, NC). Cell suspensions were then collected and centrifuged at 300 x g for 5 min. at 4°C. Red blood cells were lysed using a lysing buffer (Life Technologies, Grand Island, NY). Next, cells were washed with ice-cold 1X PBS and resuspended in 1X PBS for cell counting, which was performed using an automatic cell counter (Cellometer Auto 2000; Nexcelom Bioscience, Lawrence, MA). Lastly, dead cells were removed from the cell suspensions using the Dead Cell Removal Kit (Miltenyi Biotec) and cells were counted again using an automatic cell counter.

#### Single-cell preparation using the 10x genomics platform

Viable cells were used for single-cell RNAseq library construction using the Chromium Controller and Chromium Single Cell 3' version two kit (10x Genomics, Pleasanton, CA), following the manufacturer's instructions. Briefly, viable cell suspensions were loaded into the Chromium Controller to generate gel beads in emulsion (GEM) with each GEM containing a single cell as well as barcoded oligonucleotides. Next, the GEMs were placed in the Veriti 96-well Thermal Cycler (Thermo Fisher Scentific, Wilmington, DE) and reverse transcription was performed in each GEM (GEM-RT). After the reaction, the complementary cDNA was cleaned using Silane DynaBeads (Thermo Fisher Scentific) and the SPRIselect Reagent kit (Beckman Coulter, Indianapolis, IN). Next, the cDNAs were amplified using the Veriti 96-well Thermal Cycler and cleaned using the SPRIselect Reagent kit. Indexed sequencing libraries were then constructed using the Chromium Single Cell 3' version two kit, following the manufacturer's instructions.

#### Library preparation

cDNA was fragmented, end-repaired, and A-tailed using the Chromium Single Cell 3' version two kit, following the manufacturer's instructions. Next, adaptor ligation was performed using the

Chromium Single Cell 3' version two kit followed by post-ligation cleanup using the SPRIselect Reagent kit to obtain the final library constructs, which were then amplified using PCR. After performing a post-sample index double-sided size selection using the SPRIselect Reagent kit, the quality and quantity of the DNA were analyzed using the Agilent Bioanalyzer High Sensitivity chip (Agilent Technologies, Wilmington, DE). The Kapa DNA Quantification Kit for Illumina platforms (Kapa Biosystems, Wilmington, MA) was used to quantify the DNA libraries, following the manufacturer's instructions.

### Sequencing

Sequencing of the single-cell libraries was performed by NovoGene (Sacramento, CA) using the Illumina Platform (HiSeq X Ten System).

## Immunofluorescence

Samples of the chorioamniotic membranes, placenta villi, and decidua basal plate were embedded in Tissue-Tek Optimum Cutting Temperature (OCT) compound (Miles, Elkhart, IN) and snap-frozen in liquid nitrogen. Ten-μm-thick sections of each OCT-embedded tissue were cut using the Leica CM1950 (Leica Biosystems, Buffalo Grove, IL). Frozen slides were thawed to room temperature, fixed with 4% paraformaldehyde (Electron Microscopy Sciences, Hatfield, PA), and washed with 1X PBS. Non-specific background signals were blocked using Image-iT FX Signal Enhancer (Life Technologies) followed by blocking with antibody diluent/blocker (Perkin Elmer, Waltham, MA) for 30 min. at room temperature. Slides were then incubated with the rabbit anti-LYVE-1 antibody (Novus Biologicals, Centennial, CO) and the Flex mouse anti-human CD31 antibody (clone JC70A, Dako North America, Carpinteria, CA) for 90 min. at room temperature. Following washing with 1X PBS and blocking with 10% goat serum (SeraCare, Milford, MA), the slides were incubated with secondary goat anti-rabbit IgG–Alexa Fluor 594 (Life Technologies) and goat anti-mouse IgG–Alexa Fluor 488 (Life Technologies) for 30 min. at room temperature. Finally, the slides were washed and coverslips were mounted using ProLong Gold Antifade Mountant with DAPI (Life Technologies). Immunofluorescence was visualized using a confocal fluorescence microscope (Zeiss LSM 780; Carl Zeiss Microscopy GmbH, Jena, Germany) at the Microscopy, Imaging, and Cytometry Resources Core at the Wayne State University School of Medicine. Tile scans were performed from the chorioamniotic membranes, placental villi, and basal plate and the complete imaging fields were divided into six-by-six quadrants.

## scRNA-seq data analyses

Raw fastq files obtained from Novogene were processed using Cell Ranger version 2.1.1 from 10X Genomics. First, sequence reads for each library (sample) were aligned to the hg19 reference genome using the STAR (*Dobin et al., 2013*) aligner, and expression of gene transcripts documented in the ENSEMBL database (Build 82) were determined for each cell. Gene expression was determined by the number of unique molecular identifiers (UMI) observed per gene (QC metrics are shown in *Supplementary file 7*). Second, data were aggregated and down-sampled to take into account differences in sequencing depth across libraries using Cell Ranger Aggregate to obtain gene by cell expression data. Third, Seurat (*Butler et al., 2018*) was used to further clean and normalize the data. Then, only data from cells with a minimum of 200 detected genes, and from genes expressed in at least 10 cells were retained. Cells expressing mitochondrial genes at a level of >10% of total gene counts were also excluded, resulting in 77,906 cells and 25,803 genes (summary in *Supplementary file 1*). Gene read counts were normalized with the Seurat 'NormalizeData' function (normalization.method = LogNormalize, scale.factor = 10,000). Genes showing significant variation across cells were selected based on 'LogVMR' dispersion function and 'FindVariableGenes'. Ribosomal and mitochondrial genes were next removed, yielding 3147 highly variable genes which were subsequently analyzed using Seurat 'RunPCA' function to obtain the first 20 principal components. Clustering was done using Seurat 'FindClusters' function based on the 20 PCAs (resolution of 0.7). Visualization of the cells was performed using Uniform Manifold Approximation and Projection for Dimension Reduction (UMAP) algorithm as implemented by the Seurat 'runUMAP' function and using the first 20 principal components.

### Assigning cell type labels to single-cell clusters (Appendix 1)

Multiple methods were utilized to label the cell clusters identified by Seurat. First, marker genes showing distinct expression in individual cell clusters compared to all others were identified using the Seurat FindAllMarkers function with default parameters (*Supplementary file 3*). Marker genes with significant specificity to each cluster (*Figure 1—figure supplement 1* and *Supplementary file 3*) were compared to those reported elsewhere (*Tsang et al., 2017*; *Pavličev et al., 2017*). We also used previous known markers used by our group and https://www.proteinatlas.org/ to manually curate the labels. Further, the xCell (http://xcell.ucsf.edu/#) (*Aran et al., 2017*) tool was utilized to compare the pseudo-bulk expression signatures of the initial clusters to those of known cell types.

Additionally, we compared our manually curated cluster cell type labels to those derived from two automated cell labeling methods: SingleR (*Aran et al., 2019*) and Seurat (*Stuart et al., 2019*), using a human cell atlas reference and the placenta single-cell data in early pregnancy (*Vento-Tormo et al., 2018*) (see Appendix 1 for more details, *Figure 1—figure supplements 5–7*). Finally, we used the R package DoubletFinder (*McGinnis et al., 2019*) (https://github.com/chris-mcginnis-ucsf/DoubletFinder) to identify potential doublets. None of our clusters were impacted by doublets (*Figure 1—figure supplement 8*).

### Identification of cell-type maternal/fetal origin

We used two complementary approaches to determine the maternal/fetal origin of each cell-type. First, we used the samples derived from pregnancies where the neonate was male (3/9 cases, 8/25 samples) and we derived a fetal index based on the sum of all the reads mapping to genes on the Y chromosome relative to the total number of reads mapping to genes on the autosomes (*Figure 1—figure supplement 4*). The second method was based on genotype information derived from the scRNA-seq reads that overlap to known genetic variants from the 1000 Genomes reference panel using the freemuxlet approach implemented in popscle (*Figure 1E*). The freemuxlet approach extends the demuxlet (*Kang et al., 2018*) method, which can be useful for cases in which separate genotype information for each individual is not available. The software available at https://github.com/statgen/popscle/ was used with the '–nsample 2' option to map each cell barcode to one of the two possible genomes: fetal or maternal. The trophoblast cells are of fetal origin; therefore, we used this information to determine the fetal genome.

### Trophoblast trajectory analysis

We used the slingshot R package (*Street et al., 2018*) to reconstruct the trophoblast cell lineages from our single-cell gene expression data. This method works by building a minimum spanning tree across clusters of cells and has been reviewed as one of the most accurate tools for this task (*Saelens et al., 2019*). This analysis focused on the trophoblast cell-types (STB, CTB, EVT, and npiCTB), in which we used as input the unmerged cluster labels (i.e., four sub-clusters for CTB, and two for EVT) and the matrix of cell embedding in UMAP (see *Figure 1—figure supplement 2*).

### Differential gene expression

To identify genes differentially expressed among locations (independent of study group), we created a pseudo-bulk aggregate of all the cells of the same cell-type. Only cell types with a minimum of 100 cell in each location were considered in this analysis. Differences in cell type specific expression were estimated using negative binomial models implemented in DESeq2 (*Love et al., 2014*), including a fixed effect for each individual and location. The distribution of p-values for DEGs between pairs of compartments was assessed using a qq-plot to ensure the statistical models were well calibrated (*Supplementary file 3*). To detect DEGs across study groups we aggregated read counts across locations for each cell-type cluster, excluding cell-types with less than 100 cells in each study group (15 clusters). Differences in cell-type specific expression among study groups were estimated using negative binomial models implemented in Deseq2. Differential gene expression was inferred based on FDR adjusted p-value (q-value <0.1) and fold change >2.0.

### **Gene ontology and pathway enrichment analyses**

The clusterProfiler (*Yu et al., 2012*) package in R was utilized for the identification and visualization of enriched pathways among differentially expressed genes identified as described above. The

functions 'enrichGO', 'enrichKEGG', and 'enrichPathway' were used to identify over-represented pathways based on the Gene Ontology (GO), KEGG, and Reactome databases, respectively. Similar enrichment analyses were also conducted using Gene Set Enrichment Analysis (GSEA) (*Subramanian et al., 2005*) which does not require selection of differentially expressed genes as a first step. Significance in all enrichment analyses were based on $q < 0.05$.

## In silico quantification of single-cell signatures in maternal whole blood mRNA

### Analysis of transcriptional signatures with advancing gestation and with labor at term

Whole-blood samples collected longitudinally (12 to 40 weeks of gestation) from women with a normal pregnancy who delivered at term with (TIL) (n = 8) or without (TNL) (n = 8) spontaneous labor, were profiled using DriverMap and RNA-Seq, as previously described (*Tarca et al., 2019*) and data were available as GSE114037 dataset in the Gene Expression Omnibus. The $\log_2$ normalized read counts were averaged over the top genes (up to 20, ranked by decreasing fold change) distinguishing each cluster from all others as described above (single-cell signature). Whole blood single-cell signature expression in patients with three longitudinal samples was modeled using linear mixed-effects models with quadratic splines in order to assess the significance of changes with gestational age. Differences in single-cell signature expression associated with labor at term (TIL vs. TNL) were assessed using two-tailed equal variance t-tests. In both analyses, adjustment for multiple signature testing was performed using the false discovery rate method, with $q < 0.1$ being considered significant.

### Analysis of transcriptional signatures in preterm labor

Whole blood RNAseq gene expression profiles from samples collected at 24–34 weeks of gestation were previously described (*Paquette et al., 2018*) and data were available as GSE96083 dataset in the Gene Expression Omnibus. The study included samples from 15 women with preterm labor who delivered preterm, and 23 gestational age matched controls. $\log_2$ transformed pseudo read count data were next transformed into Z-scores based on mean and standard deviation estimated in the control group. Single cell signatures were quantified as the average of Z-scores of member genes and compared between groups using a two-tailed Wilcoxon test. Adjustment for multiple signature testing was performed using the false discovery rate method, with $q < 0.1$ being considered a significant result.

## Data and materials availability

The scRNA-seq data reported in this study has been submitted to NIH dbGAP repository (accession number phs001886.v1.p1). All other data used in this study are already available through Gene Expression Omnibus (accession identifiers GSE114037 and GSE96083) and through ArrayExpress (E-MTAB-6701). All software and R packages used herein are detailed in the Materials and methods. Scripts detailing the analyses are also available at https://github.com/piquelab/sclabor. To enable further exploration of the results we have also provided a Shiny App in Rstudio available at: http://placenta.grid.wayne.edu/.

## Additional information

### Funding

| Funder | Grant reference number | Author |
| --- | --- | --- |
| *Eunice Kennedy Shriver* National Institute of Child Health and Human Development | HHSN275201300006C | Roberto Romero |
| Wayne State University | Perinatal Research Initiative | Nardhy Gomez-Lopez Adi L Tarca |

The funders had no role in study design, data collection and interpretation.

## Author contributions
Roger Pique-Regi, Resources, Data curation, Software, Formal analysis, Supervision, Investigation, Visualization, Methodology, Project administration; Roberto Romero, Conceptualization, Resources, Supervision, Funding acquisition, Project administration; Adi L Tarca, Resources, Software, Formal analysis, Investigation, Visualization, Methodology; Edward D Sendler, Formal analysis, Investigation, Visualization; Yi Xu, Investigation, Methodology, Project administration; Valeria Garcia-Flores, Methodology, Experiments; Yaozhu Leng, Validation, Investigation, Visualization, Methodology; Francesca Luca, Resources, Methodology; Sonia S Hassan, Resources, Funding acquisition, Project administration; Nardhy Gomez-Lopez, Conceptualization, Resources, Data curation, Supervision, Funding acquisition, Validation, Investigation, Visualization, Methodology, Project administration

## Author ORCIDs
Roger Pique-Regi (iD) https://orcid.org/0000-0002-1262-2275
Roberto Romero (iD) http://orcid.org/0000-0002-4448-5121
Adi L Tarca (iD) https://orcid.org/0000-0003-1712-7588
Francesca Luca (iD) http://orcid.org/0000-0001-8252-9052
Nardhy Gomez-Lopez (iD) https://orcid.org/0000-0002-3406-5262

## Ethics
Human subjects: The collection and use of human materials for research purposes were approved by the Institutional Review Boards of the Wayne State University School of Medicine 040302M1F. All participating women provided written informed consent prior to sample collection. Data sharing certification (dbGaP phs001886.v1.p1) has been provided (see data availibility section).

## Decision letter and Author response
Decision letter https://doi.org/10.7554/eLife.52004.sa1
Author response https://doi.org/10.7554/eLife.52004.sa2

## Additional files

### Supplementary files
• Supplementary file 1. Summary of the scRNA-seq libraries prepared. Each row summarizes each 10X Genomics scRNA-seq library prepared and processed in this study: sample ID, number of cells detected after filtering, location of the tissue (BP = basal plate, PV = Placental Villi, CAM = chorioamniotic membranes), pregnancy condition (TNL = term no labor, TIL = term in labor, PTL = preterm labor), gender of the neonate, and total number of UMIs detected.

• Supplementary file 2. Summary of cell count by cell-type, location and condition. Each row summarizes the total number of cells of each cell-type as determined by Seurat and split by pregnancy condition (TNL = term no labor, TIL = term in labor, PTL = preterm labor), or location of the tissue (BP = basal plate, PV = Placental Villi, CAM = chorioamniotic membranes).

• Supplementary file 3. Marker Genes identified for each cell-type. The columns represent: 1) Cluster or cell-type name, 2) Ensembl gene identifier, 3) Gene symbol, 4) pct.1: percentage of cells in this cluster where the feature is detected, 5) pct.2: percentage of cells in other clusters where the feature is detected, 6) log fold-change of the average expression between this cluster and the rest, 7) Nominal p-value, 8) Adjusted p-value (Bonferroni).

• Supplementary file 4. Genes differentially expressed across compartments for each common cell-type. The columns represent: 1) Cluster or cell-type name, 2) Comparison groups or contrast (i.e., BP vs PV, BP vs CAM, and CAM vs PV), 3) Ensembl gene identifier, 4) Gene symbol, 5) baseMean gene baseline expression as calculated by DESeq2, 6) log2 Fold Change of the first group in column two versus the second group, 7) Standard error estimated for the log2 Fold Change, 8) Nominal p-value, 9) q-value or adjusted p-value to control for FDR. Only rows with q < 0.2 are reported.

- Supplementary file 5. Genes differentially expressed across conditions for each cell-type. The columns represent: 1) Cluster or cell-type name, 2) Comparison groups or contrast (i.e., TNL vs TIL, TIL vs PTL), 3) Ensembl gene identifier, 4) Gene symbol, 5) baseMean gene baseline expression as calculated by DESeq2, 6) log2 Fold Change of the first group in column two versus the second group, 7) Standard error estimated for the log2 Fold Change, 8) Nominal p-value, 9) q-value or adjusted p-value to control for FDR. Only rows with q < 0. two are reported.

- Supplementary file 6. Summary of the sample demographics included in this study. Data are given as medians with interquartile ranges (IQR) or as percentages (n/N). [a]One sample missing data.

- Supplementary file 7. Summary of the QC metrics for the scRNA-seq libraries prepared. Each row represents a library, and each column a QC metric reported by the 10X Cellranger software.

- Transparent reporting form

## Data availability

Protected Human subjects data deposited in dbGaP phs001886.v1.p1 Data from other sources detailed in manuscript.

The following dataset was generated:

| Author(s) | Year | Dataset title | Dataset URL | Database and Identifier |
|---|---|---|---|---|
| Pique-Regi R, Romero R, Tarca AL, Sendler ED, Xu Y, Garcia-Flores V, Leng Y, Luca F, HassanSS, Gomez-Lopez N | 2019 | Single Cell Transcriptional Signatures of the Human Placenta in Term and Preterm Parturition | https://www.ncbi.nlm.nih.gov/projects/gap/cgi-bin/study.cgi?study_id=phs001886.v1.p1 | dbGaP, phs001886.v1.p1 |

The following previously published datasets were used:

| Author(s) | Year | Dataset title | Dataset URL | Database and Identifier |
|---|---|---|---|---|
| Tarca AL, Romero R, Gomez-Lopez N, Hassan SS, Chenchik A | 2018 | Targeted sequencing based maternal whole blood expression changes with gestational age and labor in normal pregnancy | https://www.ncbi.nlm.nih.gov/geo/query/acc.cgi?acc=GSE114037 | NCBI Gene Expression Omnibus, GSE114037 |
| Paquette AG, Shynlova O, Kibschull M, Price ND, Lye SJ | 2017 | Genome Scale Analysis of miRNA and mRNA regulation during preterm labor | https://www.ncbi.nlm.nih.gov/geo/query/acc.cgi?acc=GSE96083 | NCBI Gene Expression Omnibus, GSE96083 |
| Vento-Tormo R, Efremova M, Botting RA, Turco MY, Vento-Tormo M, Meyer KB, Park JE, Stephenson E, Polański K, Goncalves A, Gardner L, Holmqvist S, Henriksson J, Zou A, Sharkey AM, Millar B, Innes B, Wood L, Wilbrey-Clark A, Payne RP, Ivarsson MA, Lisgo S, Filby A, Rowitch DH, Bulmer JN, Wright GJ, Stubbington MJT, Haniffa M, Moffett A, Teichmann SA | 2018 | Reconstructing the human first trimester fetal-maternal interface using single cell transcriptomics - 10x data | https://www.ebi.ac.uk/arrayexpress/experiments/E-MTAB-6701/ | ArrayExpress, E-MTAB-6701 |

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

## Appendix 1

## Cell type labeling procedures

Multiple methods and resources were utilized to label the clusters identified by Seurat. First, we used the function FindAllMarkers to identify the genes with significant changes in expression between each cluster and the rest of the cells using min.pct=0.33 and requiring FDR adjusted q < 10% and a $\log$(FC)> 0.5 to determine significance. Clusters with no significant differences at this threshold were merged resulting in a total of 19 clusters. For each cluster, we generated a pseudo-bulk gene expression profile and xCell (http://xcell.ucsf. edu/) (*Aran et al., 2017*) was used to compare the gene expression signatures of our clusters with those of known cell types to the default n = 64 xCell reference which includes immune cells, progenitor, epithelial, and extracellular matrix cells. Eight of the original clusters clearly identified with known cell types in the xCell reference panel that includes T-cell, B-cell, Macrophage, HSC, Fibroblast and Monocyte.

The next method we used is by comparing the marker genes identified by Seurat FindAllMarkers that passed the threshold to previously published scRNAseq marker genes (*Tsang et al., 2017*; *Pavličev et al., 2017*; *Vento-Tormo et al., 2018*) and common known markers used by our group and others https://www.proteinatlas.org/search/placenta (*Figure 1—figure supplement 1*). This resolved many of our placental (non-immune) cell clusters in the following cell types: cytotrophoblast, extravillous trophoblast, syncytiotrophoblast, decidual, endothelial, and stromal cells. To further resolve genes differentially expressed between clusters in close proximity to each other (e.g., T-cell subtypes), we ran Seurat FindMarkers function to contrast gene expression between each cluster pair, and determined as differentially expressed genes those showing a minimum logFC> 0.25 and q < 0.1. Using this analysis, we were able to label two subgroups of T cells as activated or resting. Clusters that were distinct but could not be clearly separated into well-known cell sub-types or cellular states were assigned a number (e.g., Stromal-1, Stromal-2). Some of the differences between these clusters are likely due to the maternal/fetal origin of each cell type as shown in *Figure 1B* (i.e., Macrophage 1 is likely maternal and Macrophage 2 is likely fetal) as shown by genotype analysis freemuxlet (see Materials and methods). Additionally, we used DoubletFinder https://github.com/chris-mcginnis-ucsf/DoubletFinder (*McGinnis et al., 2019*) to identify doublet cells and to ensure that none of our clusters were confounded by doublets (*Figure 1—figure supplement 8*).

Finally, we also compared our manually curated cell type identification to that derived from automated cell labeling methods SingleR (*Aran et al., 2019*) and Seurat (*Stuart et al., 2019*), (see *Figure 1—figure supplement 5* and *Figure 1—figure supplement 6*). Automated annotation provides a convenient way of transferring biological knowledge across datasets, thereby reducing the burden of interpreting clusters, but it is important to manually curate the cell labels using well established biological knowledge. If the reference database is not specific for the same tissue or similar conditions, this could lead to incorrect assignments. For SingleR, we used the vignette detailed in https://bioconductor.org/packages/devel/bioc/vignettes/ SingleR/inst/doc/SingleR.html using the human primary cell atlas (HPCA) reference provided by SingleR and the human placenta first trimester (HPFT) single cell data made available by another group (*Vento-Tormo et al., 2018*) downloaded from https://www.ebi.ac.uk/ arrayexpress/experiments/E-MTAB-6701/. For Seurat, we used only the latter reference and the standard workflow detailed in https://satijalab.org/seurat/v3.1/integration.html, and we removed any cell label with a max score > 0.001 (with almost identical results if the threshold was 0.01 or 0.0001). Similarly, we only used the pruned labels provided by SingleR.

## Lymphoid cell types

Four of our clusters correspond to lymphocyte cell-types: B-cell, NK-cell, T-cell activated, and T-cell resting. The cluster labeled as B-cell has an xCell score of 0.88 and express very highly *CD79A*. The automated labeling methods also clearly identify this cluster as B-cell when using the HPCA reference, while it was identified as Plasma cell when using the HPFT reference

panel, as no cell type is labeled as B-cell in HPFT and Plasma cell would make sense as a close match. The cluster labeled NK-cell express very highly *GNLY* and *NKG7* genes, and is also very well matched to NK-cell in the HPCA reference or one of the many NK cell types in HPFT (***Vento-Tormo et al., 2018***), which was a major focus of that study that also enriched for more rare NK cell types as they have a very important role in first trimester pregnancy, but here we only see evidence for one NK-cell cluster in (***Figure 1—figure supplement 7***). Our two clusters labeled as T-cells also closely matched the T-cell types for both reference panels and had xCell scores > 0.5, but only one T cell type is provided by those reference panels. Here, our two clusters differed in some of the genes being expressed that showed that one of the clusters was more active as indicated by signaling factors such as pro-inflammatory cytokine *TNF* and AP-1 factors such as *FOSL* and *JUNB*.

## Myeloid cell types

Three of our clusters closely matched myeloid cell types: Macrophage 1, Macrophage 2, and Monocyte. Each of these clusters closely matched to their respective cell types (xCell score > 0.8) and also when using SingleR and Seurat automated label transfer from both reference panels. Macrophage 2, which seemed to be of fetal origin, matched the Hofbauer cell type from the HPFT reference (***Vento-Tormo et al., 2018***), which are fetal resident macrophages found in the human placenta.

## Trophoblasts and other cell types

The major trophoblast cell types (CTB, EVT, and STB) expressed the markers that were defined in ***Tsang et al. (2017)***. The newly identified npiCTB also expressed the canonical CTB markers, but had a significantly higher expression of PAGE4 and decreased expression of *DDX3X*, *EIF1AX*, and *XIST* that indicate a non-proliferative state. Using automated cell labeling methods, CTB matched with VCT as defined in HPFT (***Vento-Tormo et al., 2018***), except for a small proportion that matched the SCT profile in HPFT (***Vento-Tormo et al., 2018***) (***Figure 1—figure supplement 7***). This finding may be due to differences in the expression profile of the trophoblast cells types between early and late pregnancy. The SCT in the reference panel (first trimester placental scRNA-seq data) may also include the profile of the transient stage between CTB and STB. This is supported by the trajectory analysis shown in Figure (***Figure 1—figure supplement 2***). Our EVT and STB clusters matched the labels from the automated method using the HPFT reference panel. Other small clusters showing stromal cells matched related cell types described in HPFT (***Vento-Tormo et al., 2018***).

