## [Decision Letter]

**Acceptance summary:**

The overall goal of this paper is to characterize the cell types in the placenta during different stages of labor, namely term-labor, pre-term-labor and term-no-labor by using single-cell RNA-sequencing. The novelty of this study is that they are isolating cells from different placental compartments and from women with different types of labor and stages of gestation, providing thus far the most comprehensive scRNA-seq profiling of these tissues. Overall, this study has several novel components:

a) an exceptionally novel scRNA-seq dataset, b) a set of findings that have implications in detecting pre-term labor, namely i) cell type composition varies in different placental compartments, ii) immune cell types and pathways play an important role in the timing of labor, iii) possible non-invasive monitoring of transcriptional signatures of different gestation signatures all based on maternal circulating RNA.

**Decision letter after peer review:**

Thank you for submitting your article "Single cell transcriptional signatures of the human placenta in term and preterm parturition" for consideration by *eLife*. Your article has been reviewed by three peer reviewers, one of whom is a member of our Board of Reviewing Editors, and the evaluation has been overseen by Marianne Bronner as the Senior Editor. The reviewers have opted to remain anonymous.

The reviewers have discussed the reviews with one another and the Reviewing Editor has drafted this decision to help you prepare a revised submission.

We believe this is solid and exciting work, but have a few concerns which can be addressed with writing and analyses.

Essential revisions:

1) The paper as written comes across as overly descriptive and could be improved by adding more interpretations of the findings. Many of the results are listings of a large number of cell types or genes expressed or differentially expressed among them. But it is sometimes difficult to see the overall message. For example, "Other genes highly expressed by LED cells were CD34, CDH5, EDNRB, PDPN, and TIE1 (Figure 2C, Figure 2—figure supplement 1)." Before this, it was already stated that two marker genes that define the LED cells were found. What does listing these additional genes add? Similarly, "The magnitude of cell type composition differences across placental compartments was substantial". Then there's a list of cell types and a list of genes that are highly expressed in one of the tissue. In such places it would be more impactful if there could be a more concise description along with the significance of the finding. This happens in some places, but other places could be more concise.

2) The identification of cell types was based on the expression of their clusters to those in http://xcell.ucsf.edu/#. How robust is this analysis and are all clusters annotated with a label? This is important because this is used for input for DEG between matched cell types between compartments. Also, it seems this resource is for bulk and not single cell, so please clarify how the single cell clusters were annotated with a cell label. Specifically, and very importantly, on the xCell website it is clearly stated “xCell is intended for use with bulk gene expression, for single-cell RNA-seq we recommend using SingleR: https://github.com//SingleR”. Since much of the manuscript is based on these cell/clusters annotations, it is difficult to evaluate how solid the authors claims and interpretations are (identification of new cell types, variability if cell composition amongst placental subtypes and also amongst study groups, etc) using potentially incorrectly or poorly annotated clusters.

---

## [Author Response]

Essential revisions:1) The paper as written comes across as overly descriptive and could be improved by adding more interpretations of the findings. Many of the results are listings of a large number of cell types or genes expressed or differentially expressed among them. But it is sometimes difficult to see the overall message. For example, "Other genes highly expressed by LED cells were CD34, CDH5, EDNRB, PDPN, and TIE1 (Figure 2C, Figure 2—figure supplement 1)." Before this, it was already stated that two marker genes that define the LED cells were found. What does listing these additional genes add? Similarly, "The magnitude of cell type composition differences across placental compartments was substantial". Then there's a list of cell types and a list of genes that are highly expressed in one of the tissue. In such places it would be more impactful if there could be a more concise description along with the significance of the finding. This happens in some places, but other places could be more concise.

We thank the reviewers for making this recommendation. We have revised the main text of the manuscript by incorporating new interpretation of our results and discussing our main findings within the context of previous reports. We also provide a more concise description of our findings.

You can find the changes in the Results and Discussion section.

2) The identification of cell types was based on the expression of their clusters to those in http://xcell.ucsf.edu/#. How robust is this analysis and are all clusters annotated with a label? This is important because this is used for input for DEG between matched cell types between compartments. Also, it seems this resource is for bulk and not single cell, so please clarify how the single cell clusters were annotated with a cell label. Specifically, and very importantly, on the xCell website it is clearly stated “xCell is intended for use with bulk gene expression, for single-cell RNA-seq we recommend using SingleR: https://github.com//SingleR”. Since much of the manuscript is based on these cell/clusters annotations, it is difficult to evaluate how solid the authors claims and interpretations are (identification of new cell types, variability if cell composition amongst placental subtypes and also amongst study groups, etc) using potentially incorrectly or poorly annotated clusters.

We thank the reviewers for this comment, which has greatly helped us to expand our Materials and methods section and clarify the tools used in this study. We agree with the reviewers about the importance of detailed description of the methods used to label cell types. Initially, we used xCell on the pseudo-bulk aggregate for each cluster, since there were no tools available for automated cell type labeling. We also used previous known markers used by our group and from https://www.proteinatlas.org/ to curate the labels manually. This was not clearly explained in our initial submission; thus, this section has been expanded in the revised version of the manuscript. Per the reviewers’ recommendation, we have also performed two additional analyses using SingleR and Seurat to further assess the accuracy of our initial cluster labels. Both SingleR and Seurat require a reference single cell panel to train the models to label the cells. To this end, we have used data from the human placenta in the first trimester (HPFT) (Vento-Tormo et al., 2018) as well as the human primary cell atlas (HPCA) as reference panels. We also attempted to use the data from Tsang et al., 2017 for the automated labeling; yet, the labels for each single cell were not available for this purpose. Notwithstanding, the markers from Tsang et al. were used as a guide in our original labeling. Please see Appendix 1 (“Cell type labelling procedures”) and Figure 1—figure supplement 5-7.

Overall, these additional analyses confirmed the quality of the original cluster annotation and will facilitate the comparison of our study with a single-cell RNAseq study of the placenta in early pregnancy (Vento-Tormo et al., 2018). Below, we describe the new information generated in these additional analyses.

Figure 1—figure supplement 5: Alluvial diagrams showing the correspondence between our initial curated cluster labels and automated cell-labelling methods: A) SingleR method using the HPCA reference panel; B) SingleR method using the HPFT reference panel; C) Seurat label transfer method using the HPFT reference panel.

Figure 1—figure supplement 6: Heatmap plots showing the correspondence between our initial curated clusters labels and automated cell-labelling methods: A) SingleR method using the HPCA reference panel; B) SingleR method using the HPFT reference panel; C) Seurat label transfer method using the HPFT reference panel.

These supplementary figures show the consistency among the major cell types as initially identified in our study and those obtained by the automated cell labelling methods. It is worth mentioning that this identification was limited by the cell types present in the reference panel. For example, some cell types identified in the first trimester study (NK cell subsets) were not present as separate clusters in our study (samples collected in preterm and term gestations) since the immune cell composition of the maternal-fetal interface differs between early and late gestation. Another example is the new cell type “npiCTB” identified in our study, which was not identified in the first trimester study.

Figure 1—figure supplement 7: UMAP plot showing placental single-cells. Color represents Seurat-predicted cell type labels using the HPFT reference panel.

This supplementary figure confirms that the cell types identified in our original study are in agreement with those labels derived from the automated cell labelling method using the placental single-cell study in early pregnancy. It is worth mentioning that there may be a small cluster (identified as innate lymphoid cells type 3, ILC3) that was not reported in our study since we clustered them together with T cells. ILC3s were not separately clustered in our study since they are rare and may not have distinct functions compared to T cells, which are more abundant in the placental compartments at the end of pregnancy. Moreover, a fraction of the CTBs identified in our study were labeled as SCTs (label used by Vento-Tormo et al., 2018) using automated cell labelling methods. This finding may be due to differences in the expression profile of the trophoblast cells types between early and late pregnancy. The SCT in the reference panel (first trimester placental scRNA-seq data) may also include the profile of the transient stage between CTB and STB. This is supported by the trajectory analysis shown in Figure 1—figure supplement 2.

You can find the changes in the subsection “scRNA-seq data analyses”, Appendix 1 (“Cell type labelling procedures”) and Figure 1—figure supplement 5-7.